

# Leptin-a mediates transcription of genes that participate in central endocrine and phosphatidylinositol signaling pathways in 72-hour embryonic zebrafish (*Danio rerio*)

Matthew Tuttle[1], Mark R. Dalman[2], Qin Liu[1] and Richard L. Londraville[1]

[1] Biology, University of Akron, Akron, OH, United States of America
[2] Podiatric Medicine, Kent State University, Kent, OH, United States of America

## ABSTRACT

We analyzed microarray expression data to highlight biological pathways that respond to embryonic zebrafish Leptin-a (*lepa*) signaling. Microarray expression measures for 26,046 genes were evaluated from *lepa* morpholino oligonucleotide "knockdown", recombinant Leptin-a "rescue", and uninjected control zebrafish at 72-hours post fertilization. In addition to KEGG pathway enrichment for phosphatidylinositol signaling and neuroactive ligand-receptor interactions, Gene Ontology (GO) data from *lepa* rescue zebrafish include JAK/STAT cascade, sensory perception, nervous system processes, and synaptic signaling. In the zebrafish *lepa* rescue treatment, we found changes in the expression of homologous genes that align with mammalian leptin signaling cascades including AMPK (*prkaa2*), ACC (*acacb*), $Ca^{2+}$/calmodulin-dependent kinase (*camkk2*), PI3K (*pik3r1*), Ser/Thr protein kinase B (*akt3*), neuropeptides (*agrp2*, *cart1*), mitogen-activated protein kinase (MAPK), and insulin receptor substrate (*LOC794738, LOC100537326*). Notch signaling pathway and ribosome biogenesis genes respond to knockdown of Leptin-a. Differentially expressed transcription factors in *lepa* knockdown zebrafish regulate neurogenesis, neural differentiation, and cell fate commitment. This study presents a role for zebrafish Leptin-a in influencing expression of genes that mediate phosphatidylinositol and central endocrine signaling.

# INTRODUCTION

In humans and rodents, the absence of a functional leptin (*Ob/Ob*) or leptin receptor (*Db/Db*) causes hyperphagia, early-onset and morbid obesity, severe type II diabetes, hypogonadotropic hypogonadism, impaired thermogenesis, and dysregulation of bone growth and immune response (*Friedman & Halaas, 1998*). Leptin (*LEP*), a conserved 16 kDa pleiotropic peptide hormone, is primarily secreted from adipocytes in proportion to fat mass (*Maffei et al., 1995*; *Zhang et al., 1994*). Leptin receptor (*LEPR*) has 6 isoforms in mammals that vary in cytoplasmic domain length, tissue distribution, and signaling

Corresponding author
Richard L. Londraville,
londraville@uakron.edu

competence (*Chen et al., 1996*; *Lee et al., 1996*). The leptin receptor long isoform (OB-RB) is expressed in the hypothalamus and interacts with members of the Janus kinase (JAK) and signal transducers and activators of transcription (STAT) protein families. Leptin signaling in mammals mediates anorectic (CART, POMC) and orexigenic (AgRP, NPY) circuits in the arcuate nucleus as well as voltage-gated calcium channels in the lateral hypothalamus (*Baskin et al., 1999*; *Jo et al., 2005*; *Park & Ahima, 2014*; *Schwartz et al., 1997*). *Ob/Ob*, and *Db/Db* obese rodent mutants, and diet-induced obesity mammals have reduced STAT-3 signaling in the arcuate nucleus which is generally attributed to either an absence of the hormone, an inability to respond to the hormone, or central leptin resistance (*Ghilardi et al., 1996*; *Münzberg, Flier & Bjørbæk, 2004*; *Vaisse et al., 1996*). Tissue-specific deletion of OB-RB in rodent neurons, but not hepatocytes, results in obesity, indicating that the regulatory effects of leptin signaling on adipose mass are mediated centrally (*Cohen et al., 2001*). Leptin supplementation rescues morbid phenotypes associated with *Ob/Ob* but not with *Db/Db* rodents which agrees with parabiosis experiments (*Coleman, 1973*); these findings mimic the effects of recombinant leptin in $LEP^{-/-}$ and $LEPR^{-/-}$ human case studies (*Farooqi et al., 1999*; *Farooqi et al., 2002*; *Farooqi et al., 2007*; *Gibson et al., 2004*; *Montague et al., 1997*; *Pelleymounter et al., 1995*).

Human leptin regulates JAK/STAT, AMPK/ACC, PI3K/Akt/FOXO1, and SHP2/MAPK intracellular signal transduction cascades (*Park & Ahima, 2014*). Extracellular signals transduced through leptin and insulin receptors (IR) in neuron populations of the hypothalamus stimulate IRS/PI3K/Akt/FOXO1 pathways (*Park & Ahima, 2014*; *Xu et al., 2005*). Leptin-dependent regulation of AMP-activated protein kinase (AMPK) and phosphoinositide 3-kinase (PI3K) activity in the arcuate nucleus also mediates liver homeostasis through hypothalamic-autonomic circuits in mammals (*Tanida et al., 2015*). Central leptin signaling is mitigated by negative feedback loops that involve SOCS3, which is transactivated by STAT-3, as well as PTP1B (*Dunn et al., 2005*; *Park & Ahima, 2014*). Peripheral leptin signaling in mammals activates lipid metabolism through the regulation of AMPK and acetyl coenzyme A carboxylase (*ACC*) (*Minokoshi et al., 2002*).

Are the pathways stimulated by mammalian leptin signaling conserved across vertebrates? Genomes from all vertebrate classes have at least one copy of leptin (*Londraville et al., 2017*). Despite low sequence homology, fish leptin genes share a syntenic relationship with human *lep* (*Gorissen et al., 2009*; *Howe et al., 2013*; *Kurokawa, Uji & Suzuki, 2005*), and the zebrafish genome has orthologous members of other cytokine and receptor families including IFN-I (interferon type I), IFN-II (interferon type-II), IL (interleukins), chemokines, and TNF (tumor necrosis factors) (*Savan & Sakai, 2006*). Although all mammals express single *LEP* genes, many fishes express multiple leptin paralogs (likely due to an ancestral teleost whole-genome duplication event; *Jaillon et al., 2004*). As opposed to single *LEP* orthologues in humans and rodents, the impact of having multiple copies of leptin on fish physiology and gene regulation is not well-understood. Zebrafish *lepa* message is abundant in liver, and *lepb* expression is highest in ovary (*Gorissen et al., 2009*). The putative zebrafish *lepb* promoter contains a ~1.3 kb enhancer element that responds to regeneration cues, and hepatic *lepb* expression is downregulated in response to fasting (*Gorissen et al., 2009*; *Kang et al., 2016*). Zebrafish *lepa* is induced by hypoxic cues in adult

liver and embryonic muscle, presumably through hypoxia inducible transcription factor 1 (*Chu, Li & Yu, 2010*). The induction of *lepa* and *lepb* expression in response to hypoxia and tissue regeneration, respectively, suggests that zebrafish leptins may integrate different responses to stress. The zebrafish genome has one leptin receptor (*lepr*) which has broad tissue expression including gill, liver, ovary, spleen, gut, heart, and pituitary (*Gorissen et al., 2009*; *Liu et al., 2010*). In contrast to mammalian *Db/Db* analogues, adult zebrafish leptin receptor knockouts (*lepr*$^{sa1508}$) do not have differences in adiposity, feeding, fecundity, or body size compared to controls (*Michel et al., 2016*), unlike the response to *lepr* knockout in medaka (*Chisada et al., 2014*). Knockout *lepr*$^{sa1508}$ larvae have increased numbers of $\beta$-cells as well as upregulation of insulin-a and glucagon-a (which can be blocked by metformin) suggesting that peripheral leptin signaling accommodates glucose homeostasis, but not adipostasis in zebrafish (*Michel et al., 2016*). A more complete understanding of zebrafish Leptin-a and Leptin-b binding to the receptor, and their respective signaling cascades is needed to understand leptin function in fishes.

Stat-3 signaling stimulated by leptin has been documented for several fish species (*Gong, Jönsson & Björnsson, 2016*; *Wu et al., 2016a*; *Wu et al., 2016b*; *Douros et al., 2018*) and frogs (*Cui et al., 2014*). For *Xenopus* (*Cui et al., 2014*) and *Tilapia* (*Douros et al., 2018*), large-scale transcriptomic response to leptin stimulation is established. These innovative studies add much to our understanding of the evolution of leptin response, but they do not measure transcriptomic response with reduced leptin signaling.

Zebrafish embryos (<1 hpf) injected with *lepa* morpholinos (*lepa* knockdown) are verified to have reduced Leptin-a protein expression, developmental abnormalities (reduced eye size, reduced otic vesicle formation, bent notochord), and reduced metabolic rate, all of which are ameliorated by co-injection of recombinant zebrafish Leptin-a (*lepa* 'rescue'; *Liu et al., 2012*; *Dalman et al., 2013*). In this study we used the established *lepa* knockdown model to investigate the role of Leptin-a in embryonic gene regulation. RNA prepared from *lepa* knockdown, *lepa* rescue, and control zebrafish embryos was processed on single-channel microarrays; digital expression estimates were generated for 26,046 unique gene symbol identifiers across 16 total samples. Differentially expressed genes (DEGs), KEGG pathways, and Gene Ontologies (GO) were evaluated for three pairwise comparisons: *lepa* knockdown compared to uninjected controls; *lepa* rescue compared to *lepa* knockdown; and *lepa* rescue compared to uninjected controls. We tested the null hypothesis that zebrafish Leptin-a mediates energy homeostasis through central endocrine mechanisms analogous to human and rodent leptins. We expected that the transcription of genes in adipocytokine and lipid signaling, endocrine pathways, JAK/STAT cascades, glucose homeostasis, and immune function respond to embryonic zebrafish *lepa* knockdown and recombinant Leptin-a rescue treatments.

## MATERIALS AND METHODS

### Animal care and ethical procedures

All zebrafish and associated animal procedures were reviewed and approved by The University of Akron's Institutional Animal Care and Use Committee (IACUC, approval

reference # 08-6B). Adult zebrafish were obtained from Aquatic Tropicals (Bonita Springs, FL), and maintained in aquatic fish housing systems at 28.5 °C on 13:11 light/dark cycles. Embryonic life staging, aquarium and animal maintenance procedures, diet, and husbandry approaches were completed as described in '*The Zebrafish Book*' (*Westerfield, 1995*). Age of embryo was scored as hours post fertilization (hpf) or days post fertilization (dpf). Embryos were raised at 28.5 °C using tank water supplemented with 0.01% (w/v) methylene blue (Sigma Aldrich) from 0–2 dpf.

## Microinjection

Microinjection procedures were executed as described previously (*Liu et al., 2012*). Fertilized embryos were serially collected, cleaned, and segregated from adults after spawning (<0.25 hpf). Embryos were mounted on 1.5% agarose injection plates supplemented with 0.01% (w/v) methylene blue (Sigma Aldrich). 2 nL of 0.4 mM *lepa* antisense morpholino oligonucleotides (5′-TTG AGC GGA GAG CTG GAA AAC GCA T-3′), reconstituted in Danieau buffer ([58 mM NaCl, 0.7 mM KCl, 0.4 mM MgSO$_4$, 0.6 mM Ca(NO$_3$)$_2$, 5.0 mM HEPES pH 7.6]), were delivered into 1–2 cell stage zebrafish embryos using an IM300 pneumatic microinjector (Narishige). Leptin-a "rescue" injections were prepared with 30 μM recombinant zebrafish Leptin-a (rxLeptin-a) protein stock solution (50 mM Tris, pH 8.0, >90% pure; GenScript) mixed 1:1 with 0.8 mM *lepa* morpholinos (Gene Tools), final concentration 15 μM recombinant zebrafish leptin, 0.4 mM morpholino, delivered in 2 nL. Injection needles were prepared from 0.58 mm borosilicate glass capillaries using a P-30 micropipette puller (Stoelting).

## RNA isolation and microarray processing

At 72 hpf, 5 dechorionated embryos were pooled together for each biological replicate. This was repeated 8x for uninjected control embryos (8 pools of RNA*5 embryos each = 40 total embryos), 4x for morpholino injected embryos (4 pools *5 embryos each = 20 embryos) and 4x for morpholino + recombinant leptin-injected embryos (4 pools* 5 embryos each = 20 embryos). Thus our sample size for microarray analysis was 8 control, 4 *lepa* knockdown, and 4 *lepa* rescue. RNA was isolated with TRIzol reagent (Thermo Fisher Scientific), DNase treated with the Turbo-DNA-free kit (Ambion), and washed using the RNeasy MinElute cleanup kit (Qiagen). RNA integrity (RIN) was scored with an Agilent 2100 Electrophoretic Bioanalyzer (Agilent). RNA preps were quantified using a Qubit 2.0 Fluorimeter (Thermo Fisher Scientific). Biotinylated cDNA libraries were prepared from equal amounts (250 ng) of high-quality total RNA [RIN $\geq$ 9.0], [260: 280 = 1.9 − 2.2], [260: 230 = 1.9 − 2.2] using the WT expression kit (Ambion) and the GeneChip terminal labeling kit (Ambion). After fragmentation, cDNA libraries (5.5 ug) were hybridized to single-channel Affymetrix Zebrafish 1.1 ST whole-transcriptome gene array strips (Affymetrix) for 20 h at 48 °C. Library preparation and microarray scanning procedures were performed by the University of Michigan's Microarray Core Facility following manufacturer guidelines provided for the GeneAtlas system v1.0.4.267 (Affymetrix).

 

## Microarray statistical analysis

Microarray .CEL files are deposited in ArrayExpress under accession ID E-MTAB-6548. A total of 16 .CEL files, containing probe intensities from each single-channel microarray, were placed into 'R' statistical environment v3.5.2 (*Team, 2017*) fitted with Bioconductor v3.4 (*Gentleman et al., 2004*), and associated library packages. Probeset intensities were derived from .CEL files using the Robust Multichip Average (RMA) method and 'core' summarization in 'oligo' v1.44.0 (*Bolstad et al., 2003*; *Carvalho & Irizarry, 2010*; *Irizarry et al., 2003a*; *Irizarry et al., 2003b*). Affymetrix probeset ID's were mapped to annotations from the Zv9 reference assembly with 'pd.zebgene.1.1.st' v3.12.0 (*Carvalho, 2015*), 'affycoretools' v1.52.2 (*MacDonald, 2008*), and 'org.Dr.eg.db' v3.6.0 (*Carlson, 2017*). Prior to hypothesis testing, probesets without gene symbol identifiers were filtered from the dataset. The remaining probesets were compiled to represent expression levels for single genes where each unique gene symbol identifier is represented by one probeset ranked by highest average expression across the series of arrays. Linear modeling for microarray analysis was performed on $\log_2$ probeset intensities with 'limma' v3.36.3 (*Ritchie et al., 2015*) followed by moderation of standard error using the empirical Bayesian method (*Smyth, 2005*). Three pairwise comparisons were evaluated for reliable differences in digital gene expression measures: *lepa* knockdown compared to uninjected controls; *lepa* rescue compared to *lepa* knockdown; *lepa* rescue compared to uninjected controls. Differentially expressed gene (DEG) selection criterions are adjusted p.value <0.01 and $\log_2$ fold change $<-0.5$ or >0.5 because p.value and fold change have collective merit in microarray transcriptomics (*Dalman et al., 2012*). Differentially expressed Entrez gene identifiers were mapped to zebrafish KEGG pathways ($P < 0.05$) and Gene Ontology databases ($P < 0.01$) using 'clusterProfiler' v3.8.1 (*Yu et al., 2012*) for each separate pairwise comparison. The method of Benjamini and Hochberg was used to adjust for multiple testing (*Benjamini & Hochberg, 1995*). KEGG pathway diagrams were rendered by 'pathview' v1.20.0 (*Luo & Brouwer, 2013*).

## qPCR validation of microarray results

Relative expression for 96 transcripts via qPCR was estimated in the *lepa* knockdown, *lepa* rescue, and control groups (prepared as detailed in section 2.2) using $RT^2$ Signal Transduction Pathway Finder qPCR Arrays (Qiagen). Each qPCR array was prepared in duplicate ($n = 2$ for control, knockdown, and rescue treatments) with RNA derived from 5 embryos, and equal amounts of total RNA was reverse transcribed with a qScript Flex cDNA synthesis kit (Quanta Bio) with oligo dT primers (along with no template, no primer, and no reverse transcriptase controls). Primer sequences for the qPCR assays are available from the manufacturer (cat# PAZF-014Z) including 5 reference genes (*acta 1b, b2m, hprt1, nono, rpl13a*) to which the data were normalized. Assays were run with $RT^2$ SYBR Green Master Mix (Qiagen) on an Applied Biosciences 7300 cycler. Data were analyzed with the Qiagen Gene Globe data analysis web portal (Qiagen). Within the 96 transcripts, those identified as DEG by the microarray analyses were compared across assays (qPCR vs. microarray).

## RESULTS

### Differentially Expressed Genes (DEGs)

We previously validated *lepa* morpholino knockdown using an immunoblot and mismatch morpholinos (*Liu et al., 2012*). At 72 hpf, *lepa* knockdown have reduced metabolic rate (*Dalman et al., 2013*), physical malformations pertaining to neurosensory organs (small eyes, otoliths, hindbrain), enlarged pericardial cavity and yolk sac, irregular curvature of the notochord and tail, thinning of spinal nerves, and reduced pigmentation. Injection of both rxLeptin-a and *lepa* morpholinos alleviates the *lepa* knockdown phenotype where embryos injected with higher concentrations of rxLeptin-a more closely resemble uninjected control morphology (*Liu et al., 2012*). Here, microarray expression estimates from 75,212 probesets (~1.2 million probes) were filtered to represent 26,046 unique gene symbol identifiers derived from the Zv9 reference assembly. In total, $n = 16$ microarrays were used to evaluate gene expression data derived from 40 uninjected control embryos, 20 embryos with *lepa* morpholino knockdown, and 20 embryos with *lepa* rescue; each microarray sample was derived from a pool of 5 embryos at 72 hpf.

Differentially expressed genes (DEGs) have $\log_2$ fold change $<-0.5$ or $>0.5$ and adjusted p.value $<0.01$ (Fig. 1). DEGs were rank-ordered by adjusted p.value. Genes that are differentially expressed between *lepa* knockdown and uninjected control microarray samples represent changes in gene expression that correspond to decreased Leptin-a signaling. Similarly, DEGs in the *lepa* rescue samples compared to uninjected controls present the combinatorial effect of reduced *lepa* translation (via morpholino) and induction by recombinant protein. Genes that are differentially expressed in *lepa* rescue compared to *lepa* knockdown samples respond to the induction of Leptin-a signaling (alone) because both conditions were treated with the same concentration of *lepa* morpholinos. While these microarray data alone cannot distinguish between first or second order gene targets of zebrafish Leptin-a signaling, this study is the first to measure the whole transcriptome response to reduced leptin signaling in a non-mammal.

A total of 19,987 genes were not differentially expressed in any part of the microarray dataset (76.73%). 1,461 genes were differentially expressed between *lepa* knockdown and control treatments; 425 of the 1,461 probesets were unique to this comparison. Similarly, there were 5,105 DEGs in the *lepa* rescue treatment compared to the controls; 3,125 of the 5,105 probesets were unique to this comparison. Additionally, 1,716 genes were differentially expressed in *lepa* rescue samples compared to *lepa* knockdown where 329 of the 1,716 probesets were unique to this comparison. Ultimately, 43 genes were differentially expressed among all three pairwise comparisons (Fig. 1). DEGs in each comparison are appended in the Supplementary Gene List. Microinjections, library preparation, and microarray scanning for $n = 16$ samples were executed across 5 independent trials. The sample median probeset intensities are consistent across the series of microarrays regardless of scan date or treatment (Fig. S1). Principle component analysis illustrates that the cumulative proportion of variance for principle components 1, 2, and 3 is 65.97%, or 38.77%, 18.58%, and 8.62%, respectively. *lepa* knockdown, *lepa* rescue, and uninjected control samples form segregated clusters that do not contain members from different

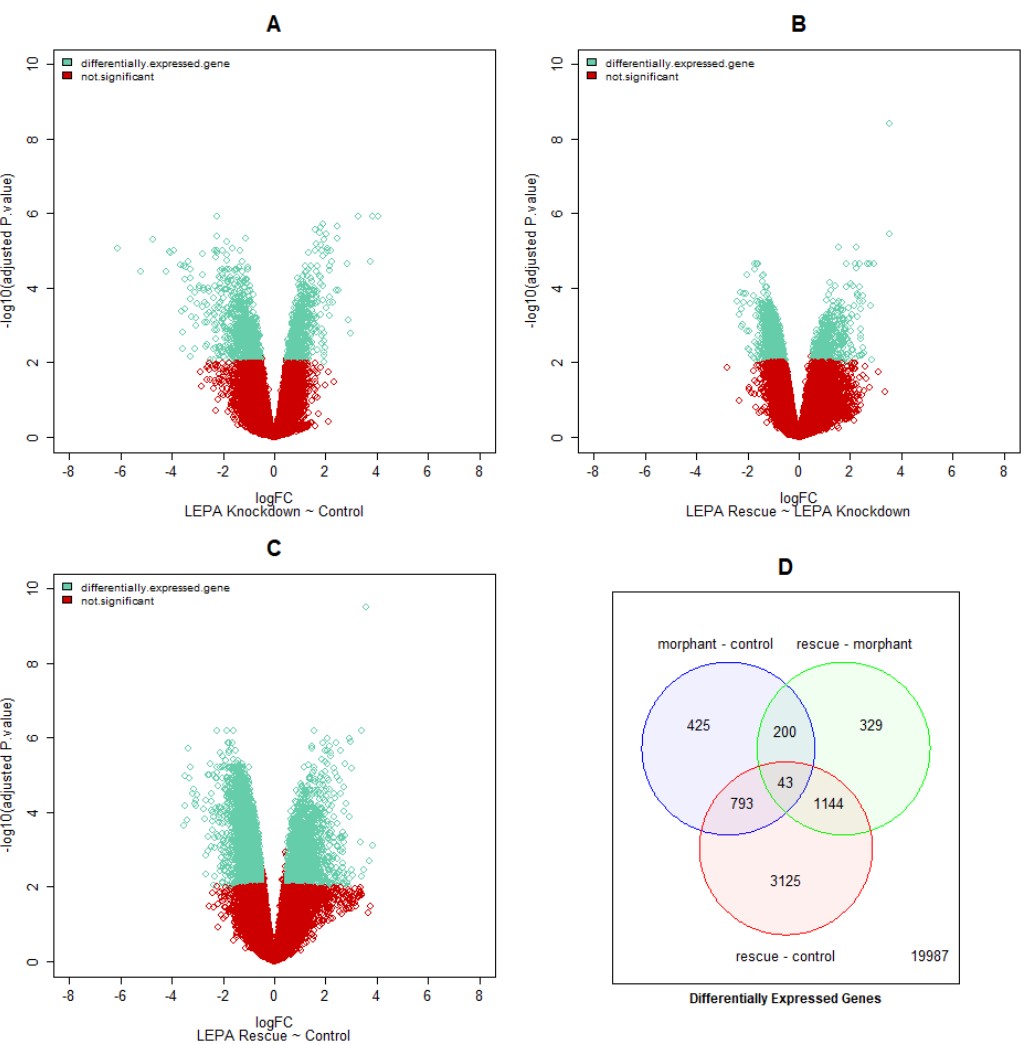

**Figure 1** **Differentially expressed genes for all datasets.** Volcano plots (1A–1C) represent the adjusted p.value versus fold change for each gene across three pairwise comparisons (listed below each plot). Each point represents one gene. Differentially expressed genes are teal; red points denote genes that do not meet differentially expressed selection criteria. Venn diagram (1D) represents the number of differentially expressed genes (DEGs) in each comparison at adjusted p.value < 0.01 and $\log_2$ fold change $\leq 0.5$ or $> 0.5$. "morphant–control" refers to genes that are differentially expressed in *lepa* knockdown zebrafish compared to uninjected controls. "rescue–morphant" refers to genes that are differentially expressed in *lepa* rescue zebrafish compared to *lepa* knockdown. "rescue–control" refers to genes that are differentially expressed in *lepa* rescue zebrafish compared to uninjected controls.

treatments, demonstrating that the most variable features in the dataset are more similar among microarrays with a common treatment as opposed to common scan date (Fig. S1).

The 50 top-ranked DEGs from each pairwise comparison are listed in Tables 1–3. KEGG and GO enrichment analyses were performed on each differentially expressed gene set (separately) to generate inferences on the regulation of the transcriptome by reduced versus induced Leptin-a signaling using knockdown and rescue treatments, respectively. Opsin 1 medium-wave-sensitive 1 (*opn1mw1*) has the largest reduction in expression of all

**Table 1** **DEG morphant vs. control.** Top-50 differentially expressed genes ranked by adjusted $p$.value from *lepa* knockdown compared to control. LogFC = Log$_2$ fold change.

| Symbol | logFC | adj.P.Val | Entrez ID |
|---|---|---|---|
| CABZ01080435.1 | 4.060502 | 1.24E-06 | NA |
| *LOC100537029* | 3.818306 | 1.24E-06 | 100537029 |
| *LOC799595* | 3.276172 | 1.24E-06 | NA |
| *igsf21b* | −2.2581 | 1.24E-06 | 567714 |
| *LOC557824* | 1.890801 | 1.92E-06 | NA |
| *nes* | 2.455737 | 2.19E-06 | 100150939 |
| *dlb* | 1.808859 | 2.32E-06 | 30141 |
| *pou2f2a* | 1.60189 | 2.70E-06 | 557055 |
| *insm1a* | 1.767329 | 3.42E-06 | 402941 |
| *notch1a* | 2.04673 | 3.61E-06 | 30718 |
| *sagb* | −2.24925 | 4.11E-06 | 619268 |
| *si:dkey-204l11.1* | 2.446131 | 4.73E-06 | 100006301 |
| *ttc7a* | −1.1399 | 4.73E-06 | 559345 |
| *LOC100537727* | 1.920483 | 5.15E-06 | NA |
| *gngt2b* | −4.75024 | 5.17E-06 | 797361 |
| *gng13b* | −1.85747 | 5.79E-06 | 436673 |
| *dld* | 1.648308 | 6.62E-06 | 30138 |
| *dnajb1b* | 1.81376 | 6.95E-06 | 327244 |
| *iars* | 2.086352 | 8.43E-06 | 334393 |
| *CTBP2* | −1.63952 | 8.78E-06 | NA |
| *opn1mw1* | −6.1018 | 8.78E-06 | 30503 |
| *gnb3b* | −3.928 | 9.69E-06 | 406483 |
| *LOC561947* | 1.846519 | 9.83E-06 | NA |
| *LOC100329434* | 1.596356 | 9.83E-06 | NA |
| *LOC568543* | −1.34136 | 9.83E-06 | NA |
| *foxn4* | 2.188575 | 1.01E-05 | 30315 |
| *dla* | 2.058014 | 1.03E-05 | 30131 |
| *LOC100536392* | −2.02991 | 1.03E-05 | NA |
| *tmx3* | −2.25409 | 1.03E-05 | 553578 |
| *si:ch211-88n13.3* | −2.31863 | 1.03E-05 | NA |
| *prph2b* | −4.08143 | 1.07E-05 | 559209 |
| *gnb3a* | −2.24428 | 1.13E-05 | 436710 |
| *brf1a* | 2.034817 | 1.17E-05 | 334402 |
| *tab1* | 1.357627 | 1.17E-05 | 403084 |
| *opn1lw2* | −4.04306 | 1.17E-05 | 436716 |
| *LOC569340* | −2.80272 | 1.22E-05 | 569340 |
| *LOC100331226* | −1.8585 | 1.29E-05 | NA |
| *neurog1* | 1.919667 | 1.38E-05 | 30239 |
| *LOC559232* | −2.30088 | 1.48E-05 | NA |
| *cacna2d4b* | −1.85537 | 1.82E-05 | 100150428 |
**Table 1** (*continued*)

| Symbol | logFC | adj.P.Val | Entrez ID |
|--------|-------|-----------|-----------|
| *LOC570404* | 3.754077 | 1.97E-05 | NA |
| *LOC562934* | −1.23679 | 2.01E-05 | NA |
| *arr3a* | −3.38027 | 2.01E-05 | 436678 |
| *nelfa* | 2.257915 | 2.03E-05 | 559677 |
| *hbbe3* | 2.846799 | 2.26E-05 | 30596 |
| *zgc:73359* | −3.65251 | 2.37E-05 | 393810 |
| *cxcl-c1c* | 2.146952 | 2.51E-05 | NA |
| *arhgef1b* | 1.687267 | 2.51E-05 | 557983 |
| *si:ch1073-303l5.1* | 1.40302 | 2.51E-05 | NA |
| *lin28a* | 1.191522 | 2.51E-05 | 394066 |

genes in both *lepa* knockdown (−6.10 $\log_2$ fold change) and *lepa* rescue (−3.82 $\log_2$ fold change) treatments compared to control. In the *lepa* knockdown and control contrast, the top-5 ranked genes include immunoglobulin superfamily DCC subclass member 3-like (*LOC100537029,* 3.27 $\log_2$ fold change) and immunoglobin superfamily member 21b (*igsf21b,* −2.25 $\log_2$ fold change). Von Willebrand factor (*vwf;* 3.55 $\log_2$ fold change) and SI:DKEY-24I24.3 (3.52 $\log_2$ fold change) are the two highest-ranked genes in the *lepa* rescue treatment compared to the controls. In *lepa* rescue zebrafish, leptin-b (*lepb*) is upregulated (1.76 $\log_2$ fold change) compared to controls. In *lepa* knockdown zebrafish, *lepa* and *lepr* are differentially expressed (1.01 and 0.77 $\log_2$ fold changes, respectively) compared to controls which may be compensatory to *lepa* knockdown.

## KEGG pathway enrichment

To avoid the inherent problems of multiple comparisons with transcriptomics studies, data reducing techniques that identify classes of genes and pathways that are differentially expressed are common and useful ways of analyzing these datasets. Gene Ontologies (GO) are classifiers used to group gene sets by similarity in function based on phylogenetic, computational, and experimental inferences. GO categories are segregated into biological process (BP), cellular component (CC), and molecular function (MF) components. Similarly, Kyoto Encyclopedia of Genes and Genomes (KEGG) pathway analysis is another classifier that maps interactions within a gene set to coordinated biochemical signaling pathways and molecular functions. KEGG pathway results ($P < 0.05$) are summarized in Table 4 for all three pairwise comparisons; DEGs that map to each enriched KEGG pathway and GO category are appended in the Supplementary Gene List.

Notch signaling is the only differentially expressed KEGG pathway in *lepa* rescue zebrafish compared to *lepa* knockdown (represented by DEGs *dlb, notch1a, notch1b, ep300a, dld, notch3, jag1a, dlc, crebbpb, dvl3a, dla, kat2a*) (Table 4). The highest-ranked KEGG pathway in *lepa* knockdown compared to control zebrafish is phototransduction (*rho, gnat2, rgs9a, grk7a, grk1b, rcvrna, gnat1, saga, pde6a, pde6b, calm1a, calm1b, guca1d, grk1a*), followed by ribosome biogenesis in eukaryotes (*nmd3, tbl3, rexo1, rpp25l, mphosph10, rcl1, gar1, utp18, pwp2 h, dkc1, wdr3, mdn1*), notch signaling (*dlb, notch1a,*

**Table 2 Top DEG rescue vs. knockdown.** Top-50 differentially expressed genes ranked by adjusted p.value from *lepa* rescue compared to *lepa* knockdown. LogFC = Log$_2$ fold change.

| Symbol | logFC | adj.P.Val | Entrez ID |
|---|---|---|---|
| *SI:DKEY-24I24.3* | 3.551056 | 3.87E-09 | NA |
| *vwf* | 3.524634 | 3.49E-06 | 570643 |
| *LOC568400* | 2.24809 | 8.17E-06 | NA |
| *LOC570208* | 1.540514 | 8.19E-06 | NA |
| *or125-2* | 2.928568 | 2.24E-05 | 100150140 |
| *LOC793072* | 2.752483 | 2.24E-05 | NA |
| *or115-6* | 2.690795 | 2.24E-05 | 678539 |
| *LOC100535281* | 2.679196 | 2.24E-05 | NA |
| *si:ch211-237a4.2* | 2.352812 | 2.24E-05 | 100034537 |
| *si:dkey-28d5.10* | 2.216788 | 2.24E-05 | 799800 |
| *or125-4* | 1.839774 | 2.24E-05 | 100148706 |
| *cdk11b* | −1.58101 | 2.24E-05 | 494103 |
| *LOC100333199* | −1.59951 | 2.24E-05 | NA |
| *SOS1* | −1.61741 | 2.24E-05 | NA |
| *plxnd1* | −1.7363 | 2.24E-05 | 402998 |
| *IGHV2-2* | 2.245677 | 2.89E-05 | NA |
| *LOC794788* | 2.231972 | 2.89E-05 | NA |
| *LOC100333311* | −1.72142 | 3.12E-05 | NA |
| *depdc1b* | −2.00778 | 4.36E-05 | 100006170 |
| *zgc:158376* | −1.36533 | 4.52E-05 | 100101643 |
| *dph6* | −1.42828 | 4.73E-05 | 503603 |
| *bcl2* | −1.40772 | 6.54E-05 | NA |
| *galnt10* | −1.42208 | 6.54E-05 | 767665 |
| *LOC100007376* | 1.217926 | 7.14E-05 | 100007376 |
| *LOC570185* | 2.012359 | 7.71E-05 | 570185 |
| *adrb1* | 2.119886 | 8.64E-05 | 557194 |
| *si:ch211-160j14.2* | 1.438866 | 8.88E-05 | 792922 |
| *LOC799771* | −1.23956 | 9.23E-05 | NA |
| *LOC100534901* | 2.382595 | 9.48E-05 | NA |
| *LOC100536834* | 1.848393 | 9.48E-05 | 100536834 |
| *pou2f2a* | −1.3119 | 9.48E-05 | 557055 |
| BX936386.1 | 1.247975 | 0.000113 | NA |
| *slitrk5* | −1.37301 | 0.000137 | 100330023 |
| *LOC100149966* | −2.27763 | 0.000137 | 100149966 |
| *taar10c* | 2.26676 | 0.000138 | 794440 |
| *cdc42bpb* | −1.18873 | 0.000138 | 567039 |
| *si:rp71-56i13.6* | −2.08428 | 0.000138 | 564457 |
| *USP30* | −2.15151 | 0.000141 | NA |
| *or106-2* | 1.17958 | 0.000148 | 100861459 |
| *LOC100536940* | 2.318071 | 0.000154 | NA |

**Table 2** (*continued*)

| Symbol | logFC | adj.P.Val | Entrez ID |
|---|---|---|---|
| CXCL11 | 2.504981 | 0.000162 | NA |
| zgc:85975 | −1.77849 | 0.000166 | 406549 |
| fzd5 | −1.2611 | 0.000167 | 30364 |
| LOC100535489 | 2.303531 | 0.000167 | NA |
| polh | −1.14961 | 0.000203 | 678520 |
| SI:DKEY-256I11.6 | 1.098104 | 0.000217 | NA |
| zbtb16a | −1.49196 | 0.000217 | 323269 |
| LOC100331199 | 2.513658 | 0.000231 | NA |
| LOC560659 | 2.383395 | 0.000231 | 560659 |
| trim35-3 | 1.341874 | 0.000231 | 100003935 |

*dld, dla, notch1b, notch3, jag1a, her15.1, dlc, rbpjl*), then neuroactive ligand–receptor inter-action (*grin1b, gabrd, grm1a, grin2ab, drd4b, gria3a, grm3, grin1a, gabrb2, gabbr2, chrna6, lepa, si:dkey-1h24.2, oxtr, thrab, LOC100330554, grik1a, crhr1, drd1b, vipr2, chrm4a, calcr, oprm1, grm6b, LOC562831, vipr1b, lepr, gria1b, p2rx2, LOC100334689, gria2b, glra3, grm6a, gnrhr4*). *lepa* rescue compared to control samples identify neuroactive ligand–receptor interaction (94 DEGs; Fig. 2) and phototransduction (*gnat2, rho, grk7a, saga, gnat1, grk1a, rcvrna, rgs9a, calm1a, calm1b, exorh, grk1b, rgs9b*) similar to *lepa* knockdown. However, rescue vs. control also identifies phosphatidylinositol signaling (Fig. 3; *mtmr1a, ip6k1, LOC567728, itpr2, mtmr7b, inpp5e, mtmr3, plcd3a, plcb4, pip5k1ca, ipmkb, itpr1a, pi4kaa, calm1a, dgkh, LOC100333801, dgkza, plcg2, pik3r1, prkcba, calm1b, ppip5k2, synj1, prkcbb, inpp5f, mtmr6, plcg1, dgke, itpkca*) and glycosaminoglycan biosynthesis (*xylt1, ext1c, ndst3, hs3st1l1, hs6st2, glcea, hs3st2, hs6st1a, extl3, glceb, hs3st3b1b*).

The top-50 DEGs in *lepa* knockdown compared to controls are clustered to illustrate expression patterns among genes and microarray samples (Fig. 4A). All microarray samples clustered with members from the same treatment. The heatmap includes 5 transcription factors (*neurog1, pou2f2a, insm1a, brf1a, foxn4*) as well as components of the Delta/Notch and phototransduction genes (demarcated by orange and purple, respectively). Phototransduction and delta/notch genes display reciprocal trends in expression with respect to treatment (down with knockdown, up with rescue) indicating that they may be coregulated. Delta/Notch has high expression in *lepa* knockdown and low expression in both the control and *lepa* rescue samples. Conversely, phototransduction genes have high expression in controls but low expression in *lepa* knockdown and rescue treatments. Figure 4B clusters homologous gene targets that are regulated by the mammalian leptin signaling pathway (*lepa, lepb, lepr, LOC794738, LOC100537326, akt3, pik3r1, prkaa2, socs2, socs5b, socs9, npy8br, npy8ar, mc5ra, foxo1b, jak2a, jak2b, camkk2, map3k12, agrp2, cart1*). These targets are as identified in mammals by *Park & Ahima (2014)*. Figure 4B also documents genes displaying large fold changes (*vwf, opn1lw1, igsf21b, LOC100537029*), and others that regulate lipid (*acacb, dagla, daglb, lpl, crot, plin1, cpt1b, pank2*) or RNA metabolism (smg5, dicer1, *LOC796505, LOC570775*).

**Table 3 Top DEG rescue vs. control.** Top-50 differentially expressed genes ranked by adjusted p.value from *lepa* rescue compared to control. LogFC = Log$_2$ fold change.

| Symbol | logFC | adj.P.Val | Entrez ID |
|--------|-------|-----------|-----------|
| SI:DKEY-24I24.3 | 3.599239 | 3.26E-10 | NA |
| vwf | 3.385689 | 6.70E-07 | 570643 |
| LOC796392 | −1.60622 | 6.70E-07 | NA |
| LOC570208 | 1.538632 | 6.75E-07 | NA |
| LOC100149028 | −1.87278 | 6.75E-07 | NA |
| pank2 | −2.25072 | 6.75E-07 | 570866 |
| LOC793072 | 2.935593 | 1.06E-06 | NA |
| or115-6 | 2.933138 | 1.06E-06 | 678539 |
| LOC568400 | 2.075418 | 1.06E-06 | NA |
| or125-2 | 2.962312 | 1.34E-06 | 100150140 |
| or125-4 | 1.906985 | 1.40E-06 | 100148706 |
| parp6a | −1.64685 | 1.40E-06 | 436810 |
| si:ch211-243a15.1 | −1.77679 | 1.40E-06 | 100005033 |
| map3k12 | −1.82325 | 1.40E-06 | 404626 |
| IGHV2-2 | 2.409117 | 1.42E-06 | NA |
| LOC100536180 | 2.170446 | 1.66E-06 | NA |
| SI:DKEY-256I11.6 | 1.448259 | 1.66E-06 | NA |
| slitrk5 | −1.67491 | 1.93E-06 | 100330023 |
| LOC569340 | −3.34108 | 1.93E-06 | 569340 |
| BX936386.1 | 1.488632 | 1.97E-06 | NA |
| LOC100535281 | 2.606151 | 2.21E-06 | NA |
| si:dkey-28d5.10 | 2.068829 | 2.21E-06 | 799800 |
| TMPO | 1.986495 | 2.21E-06 | NA |
| LOC562934 | −1.51774 | 2.21E-06 | NA |
| LOC568961 | −1.74594 | 2.21E-06 | NA |
| LOC794788 | 2.211161 | 2.56E-06 | NA |
| CTBP2 | −1.74604 | 2.56E-06 | NA |
| LOC100332615 | −2.10437 | 2.56E-06 | 100332615 |
| si:ch211-237a4.2 | 2.20162 | 3.12E-06 | 100034537 |
| cdh23 | −2.03904 | 3.20E-06 | 407978 |
| LOC100536867 | 1.858425 | 3.81E-06 | 100536867 |
| sc:d0343 | −1.59675 | 3.81E-06 | NA |
| LOC558743 | −1.42439 | 4.01E-06 | NA |
| zgc:123060 | −1.89975 | 4.07E-06 | 641487 |
| LOC100536834 | 1.974369 | 4.33E-06 | 100536834 |
| LOC570185 | 2.056828 | 4.91E-06 | 570185 |
| ankrd13b | −1.22985 | 4.91E-06 | 568981 |
| lrrtm1 | −1.26398 | 4.91E-06 | 570385 |
| arvcfa | −1.43558 | 4.91E-06 | 572216 |
| ddr2 | −1.55141 | 4.91E-06 | NA |

**Table 3** (*continued*)

| Symbol | logFC | adj.P.Val | Entrez ID |
|---|---|---|---|
| *lrit1b* | −2.51937 | 4.91E-06 | 553432 |
| *LOC560659* | 2.755338 | 5.11E-06 | 560659 |
| *LOC100006857* | −1.38684 | 5.45E-06 | NA |
| *lin28a* | 1.338285 | 5.56E-06 | 394066 |
| *sagb* | −1.99538 | 5.65E-06 | 619268 |
| *igsf21b* | −1.70889 | 6.12E-06 | 567714 |
| *LOC555422* | −2.11608 | 6.12E-06 | 555422 |
| *nr1d1* | −3.2824 | 6.12E-06 | 494487 |
| *LOC100333199* | −1.33657 | 6.14E-06 | NA |
| *zgc:111992* | 1.544128 | 6.26E-06 | NA |

**Table 4  KEGG pathway enrichment all pairwise comparisons.** Kyoto Encyclopedia of Genes and Genomes (KEGG) pathway enrichment ($P < 0.05$) for 3 separate pairwise comparisons: I. *lepa* knockdown compared to control (MO::WT), II. *lepa* rescue compared to *lepa* knockdown (RS::MO), and III. *lepa* rescue compared to control (RS::WT).

| | KEGG Pathway (<0.05); LEPA Rescue—Control (3,036 Entrez ID's; 5,105 DEGs) | Adj. *p*.value | DEG Count |
|---|---|---|---|
| dre04080 | Neuroactive ligand–receptor interaction | 4.63E-07 | 94 |
| dre04070 | Phosphatidylinositol signaling system | 2.83E-03 | 29 |
| dre04744 | Phototransduction | 9.92E-03 | 13 |
| dre00534 | Glycosaminoglycan biosynthesis - heparan sulfate/heparin | 1.50E-02 | 11 |
| | KEGG Pathway (<0.05); LEPA Rescue—LEPA knockdown (999 Entrez gene ID's) | Adj. *p*.value | DEG Count |
| dre04330 | Notch signaling pathway | 0.000108 | 12 |
| | KEGG Pathway (<0.05); LEPA knockdown—Control (970 Entrez ID's) | Adj. *p*.value | DEG Count |
| dre04744 | Phototransduction | 2.03E-08 | 14 |
| dre03008 | Ribosome biogenesis in eukaryotes | 6.65E-03 | 12 |
| dre04330 | Notch signaling pathway | 1.08E-02 | 10 |
| dre04080 | Neuroactive ligand–receptor interaction | 2.16E-02 | 34 |

## Gene ontology enrichment

The *lepa* knockdown compared to control GO data were generated from 1,461 DEGs (970 Entrez gene ID's); *lepa* knockdown have 54 enriched biological process terms ($P < 0.01$) including visual perception, ncRNA metabolic process, rRNA metabolic process, ribosomal large/small subunit biosynthesis, neurological system process, protein-chromophore linkage, spinal cord development, nervous system development and neuron differentiation, regulation of neurotransmitter levels and secretion, synaptic signaling, and regulation of neurogenesis (Fig. 5). The GO data for the *lepa* rescue and knockdown contrast were generated from 1,716 DEGs (999 Entrez gene ID's) that correspond to 22 enriched terms including sensory perception, Notch signaling, brain/head/gland/ventral spinal cord development, regionalization, olfactory and semaphorin receptor activity, and protein tyrosine kinase activity (Fig. 6). Biological process Gene Ontologies from the *lepa* rescue

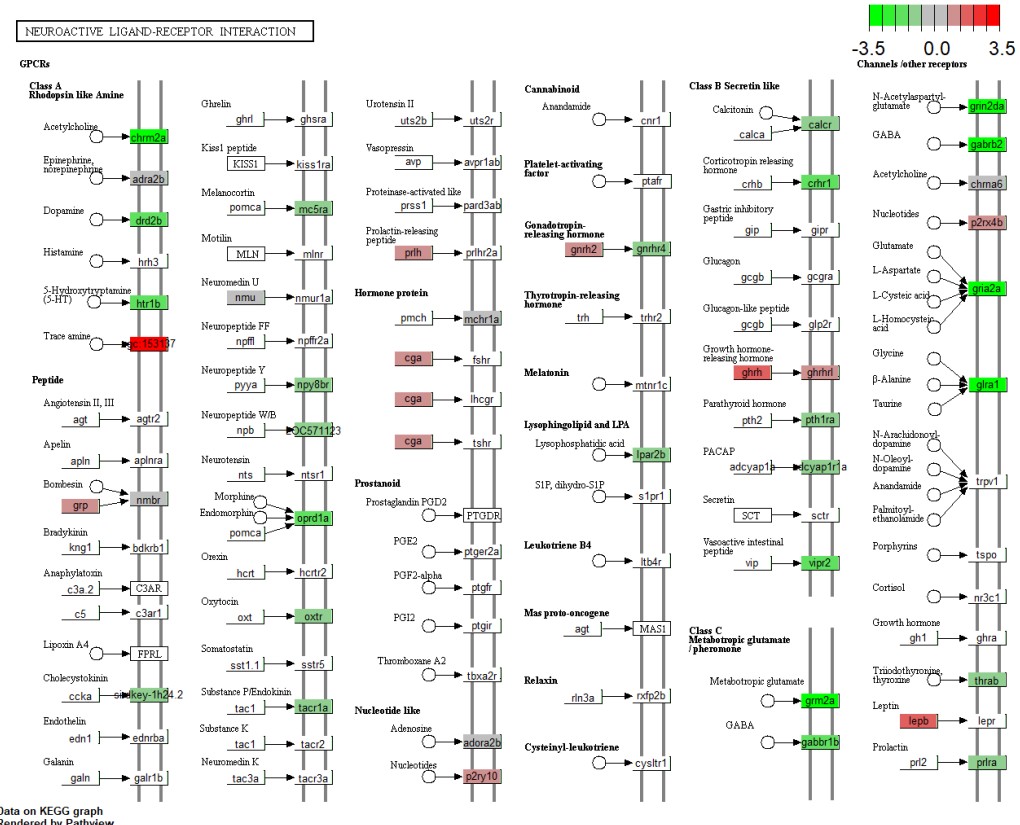

**Figure 2 Neuractive Ligand-Receptor Interactions.** (Kyoto Encyclopedia of Genes and Genomes (KEGG)) pathway enrichment ($P < 0.05$) representing neuroactive ligand-receptor interactions from *lepa* rescue compared to control microarrays. Color scale reflects $\log_2$ fold change for differentially expressed genes.

and control comparison were produced using 5,105 DEGs (3,036 Entrez gene IDs); *lepa* rescue has 39 enriched biological process terms including JAK-STAT cascade, nervous system process, synaptic signaling, synapse organization, sensory perception, regulation of voltage-gated calcium channels, neurotransmitter secretion, and phototransduction (Fig. 7). Genes that map to each GO category in Figs. 5–7 are appended in the Supplementary Gene List.

From 1,461 DEGs in *lepa* knockdown compared to controls, there were 112 transcription factors (7.67%) (Fig. S2). There are 5 differentially expressed transcription factors with a $\log_2$fold change greater than 2 (*foxn4, fosl1a, cbx7a, and atf5a*). Two transcription factors (*pou2f2a, insm1a*) ranked in the top-10 of all 1,461 DEGs between *lepa* knockdown and controls (Table 1). Additionally, the Gene Ontology enrichment analysis for transcription factor biological process has 65 terms that envelop many aspects of development (spinal cord, endocrine system, pancreas, brain, head, neural retina, nervous system, epithelium), as well as regulation of neurogenesis, neuron differentiation, cell fate commitment, and hindbrain morphogenesis (Fig. S3).

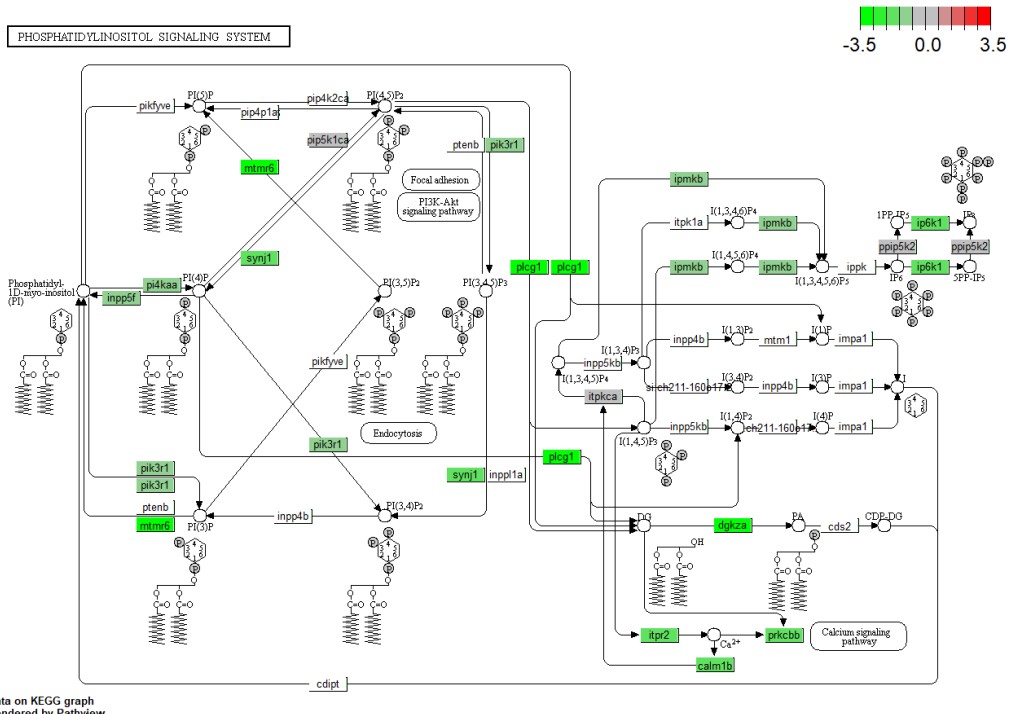

**Figure 3 Phosphatidyl Inositol Signaling.** Kyoto Encyclopedia of Genes and Genomes (KEGG) pathway enrichment ($P < 0.05$) representing phosphatidylinositol signaling from *lepa* rescue compared to control microarrays. Color scale reflects $\log_2$ fold change for differentially expressed genes.

## qPCR validation of the microarray

Several genes within the qPCR dataset were also identified as DEG in the microarray experiments (10 for knockdown vs. control and 5 for rescue vs. control; Fig. S4). Fold differences >1 are positive changes in expression, and <1 are negative. The direction of expression change is the same between microarray and qPCR experiments for 8 out of 10 genes for the comparison of knockdown to control. For the rescue vs. control comparison, 1 out of 5 genes has the same direction of change when expression is measured by microarray vs. qPCR.

## DISCUSSION

Null mutations in leptin (*LEP*) or leptin receptor (*LEPR*) impair lipid metabolism, endocrine signaling, and energy homeostasis in humans and rodents, but there are few non-mammal model organisms with reduced leptin signaling. Mammalian leptin is widely characterized in adults, and to a lesser extent in juveniles or embryos (*Antczak & Van Blerkom, 1997*). Embryonic leptin expression precludes terminal adipocyte differentiation in mammals suggesting that leptin modulates embryogenesis apart from its role as an adipocytokine (*Antczak & Van Blerkom, 1997*). Given the intrinsic limitations of *in vivo* embryonic leptin manipulation using mammals, we tested transcriptomic response to leptin manipulation in developing zebrafish using *lepa* morpholino knockdown. Our

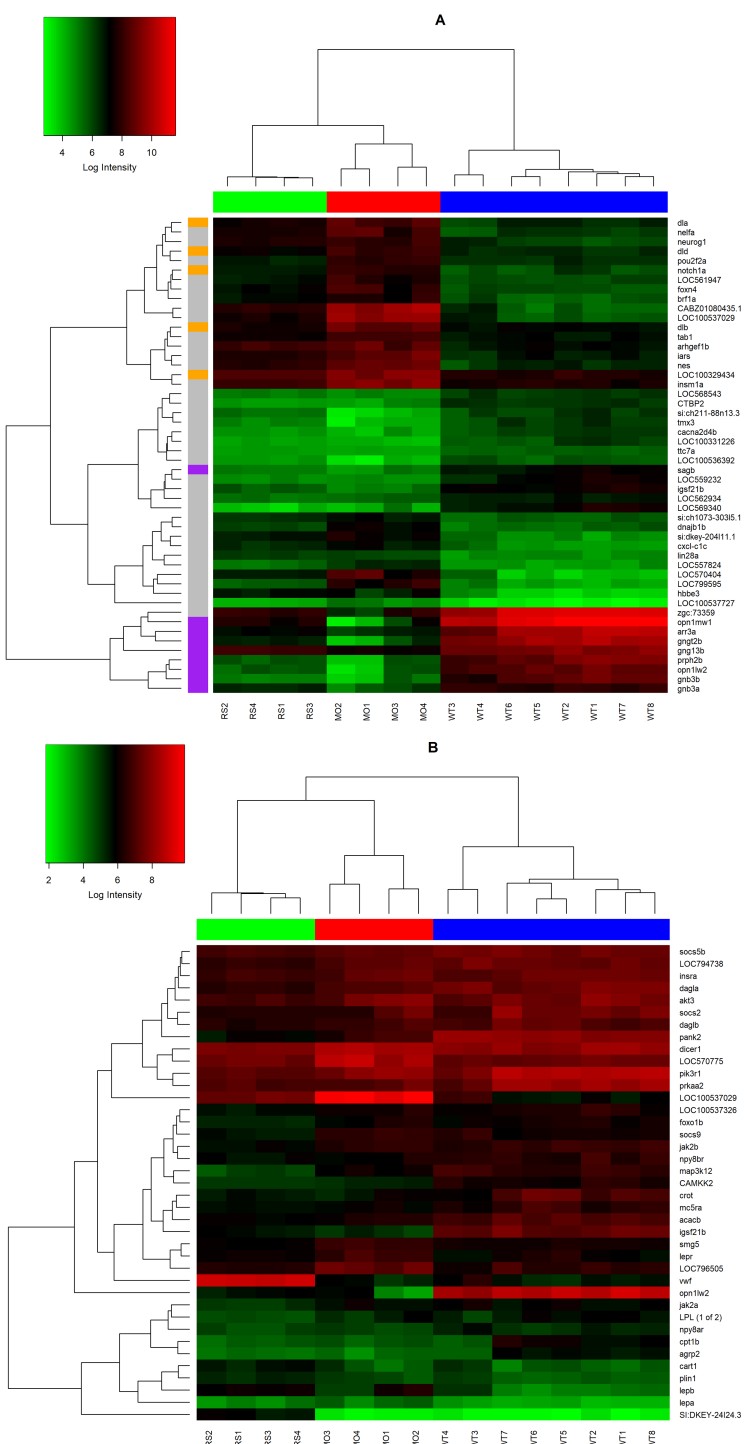

**Figure 4  Top-Ranked DEGs (A) and Leptin Signaling DEGs.** (A) Heatmap displaying the top-50 ranked DEGs in *lepa* knockdown compared to controls. Genes (rows) are clustered to represent Delta/Notch genes (orange), Phototransduction (purple), or grey (other). (B) Heatmap containing the zebrafish homologs of leptin signaling genes identified by *Park & Ahima (2014)*, including *lepa, lepb, lepr, LOC794738, LOC100537326, akt3, pik3r1, prkaa2, socs2, socs5b, socs9, npy8br, npy8ar, mc5ra, foxo1b, jak2a, jak2b, camkk2, map3k12, agrp2,* (continued on next page...)

**Figure 4 (…continued)**
and *cart1*. Genes displaying large fold changes (*vwf, opn1lw1, igsf21b, LOC100537029*), and others that regulate lipid (*acacb, dagla, daglb, lpl, crot, plin1, cpt1b, pank2*) or RNA metabolism (smg5, dicer1, *LOC796505, LOC570775*) are also included. Each gene (row) is color coded to represent high (red) and low (green) levels of expression. Columns are color coded by treatment—blue (*lepa* rescue), red (*lepa* knockdown), green (uninjected controls).

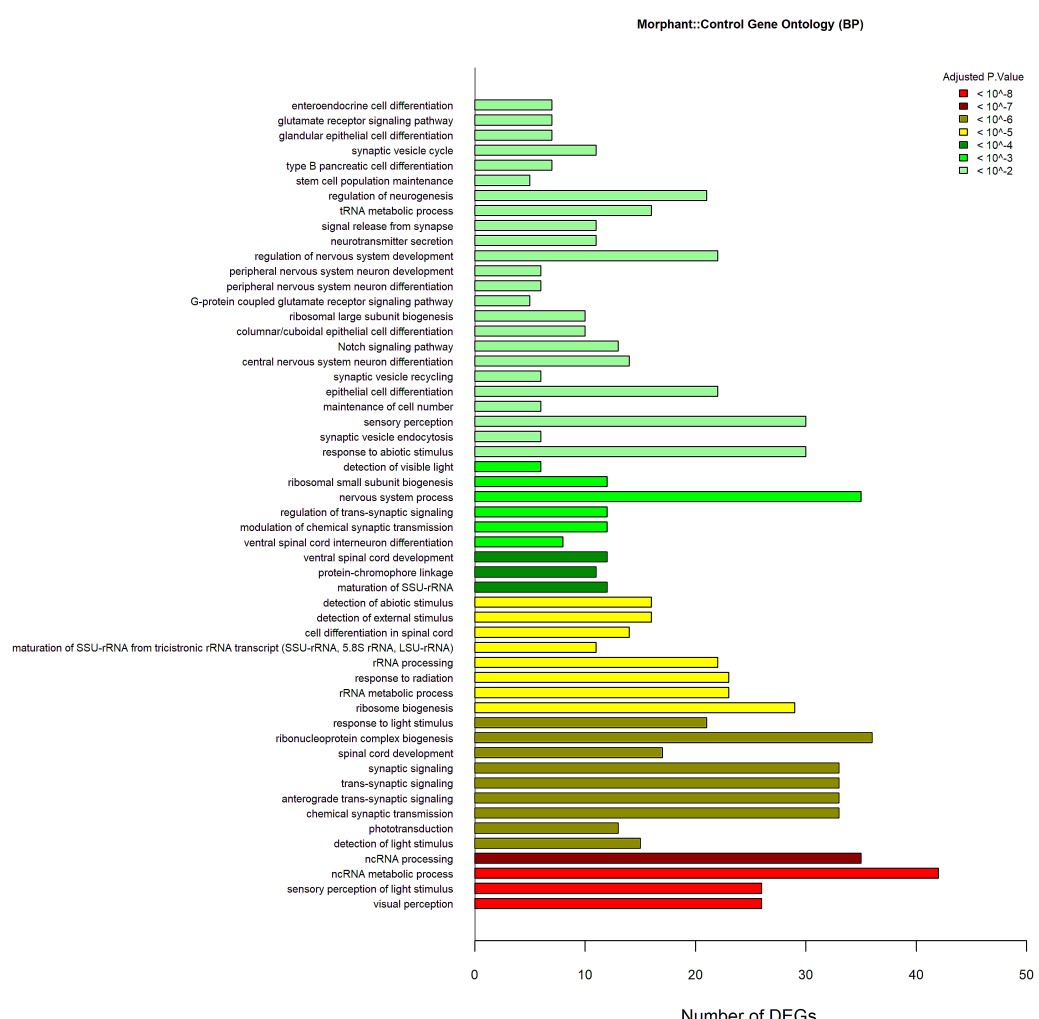

**Figure 5  GO Analyses knockdown Vs. Control.** Gene Ontology biological process enrichment results ($P < 0.01$) from *lepa* knockdown compared to control; the barplot contains 54 enriched terms. Color scale reflects adjusted *p*.value; the *x*-axis denotes the number of DEGs that map to each enriched term.

findings highlight roles for Leptin-a tied to the regulation of central endocrine and phosphatidylinositol signaling gene expression in early zebrafish development which align with canonical targets of leptin signal transduction cascades in mammals.

Zebrafish *lepa* and *lepb* are co-expressed in zebrafish pituitary which may mediate endocrine and local (paracrine, autocrine) axes, respectively (*Gorissen et al., 2009*). Affirming a role for Leptin-a signaling in the central nervous system, neuroactive

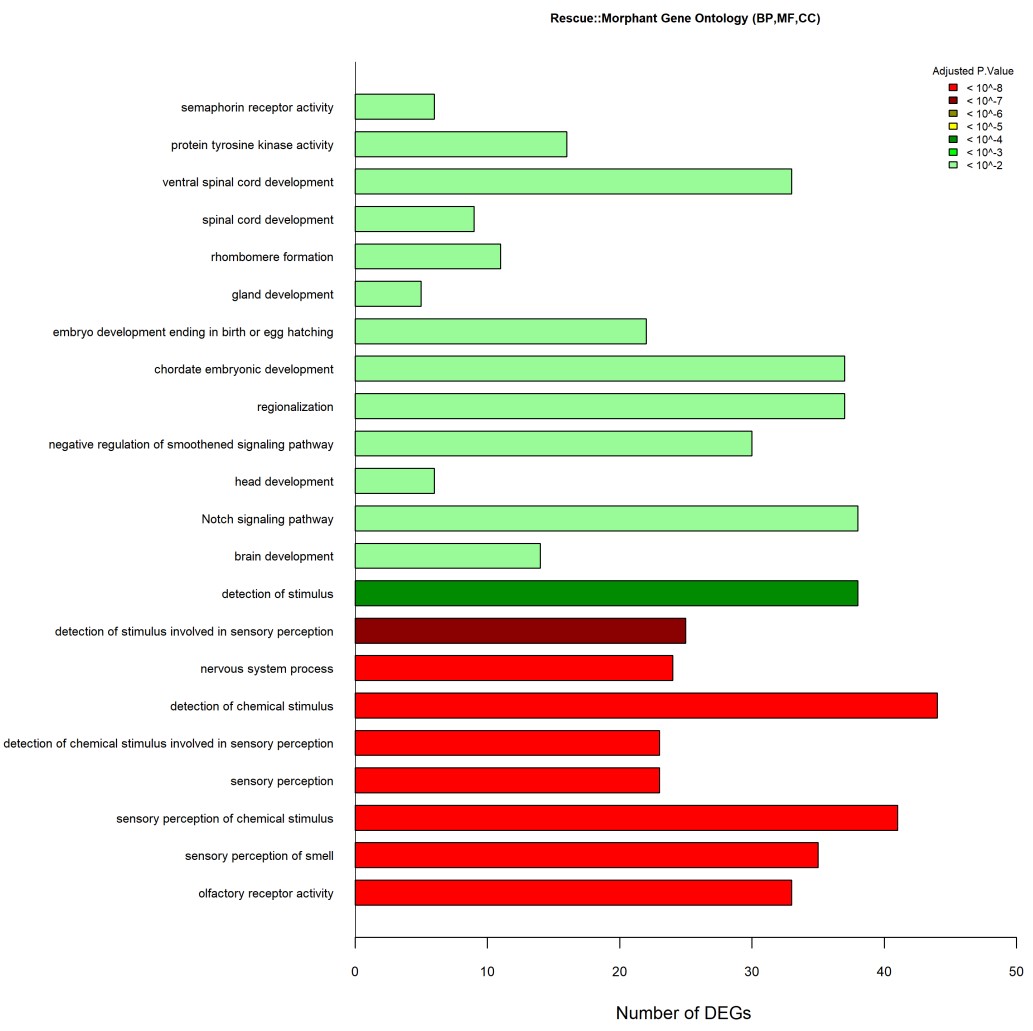

**Figure 6** **GO Analyses Rescue Vs. knockdown.** Gene Ontology enrichment results ($P < 0.01$) containing $n = 22$ terms derived from all three categories (biological process, cellular component, and molecular function) in *lepa* rescue compared to *lepa* knockdown treatments. Color scale reflects adjusted *p*.value; the *x*-axis denotes the number of DEGs that map to each enriched term.

ligand–receptor interactions are enriched KEGG pathways in *lepa* knockdown and rescue zebrafish compared to controls, but fewer genes respond to knockdown as opposed to rescue suggesting there are differential responses to reduced versus induced Leptin-a signaling (Fig. 2). Additionally, *lepa* rescue DEGs are analogous to neuroendocrine factors associated with the human leptin pathway (and classical mediators of changes in appetite and metabolism), including agouti-related peptide-2 (*agrp2*), cocaine-and-amphetamine-regulated transcript 1 (*cart1*), neuropeptide Y receptor Y8 (*npy8ar, npy8br*), and melanocortin receptor 5a (*mc5ra*) (Fig. 4B). Zebrafish homologs of the human *lep/pi3k/akt*, and *lep/shp2/mapk* cascades are differentially regulated in *lepa* rescue (but not *lepa* knockdown) treatments including protein kinase B (*akt3*), phosphatidylinositol 3-kinase (*pik3r1*), insulin receptor substrate (*LOC794738, LOC100537326*), Janus kinase 2

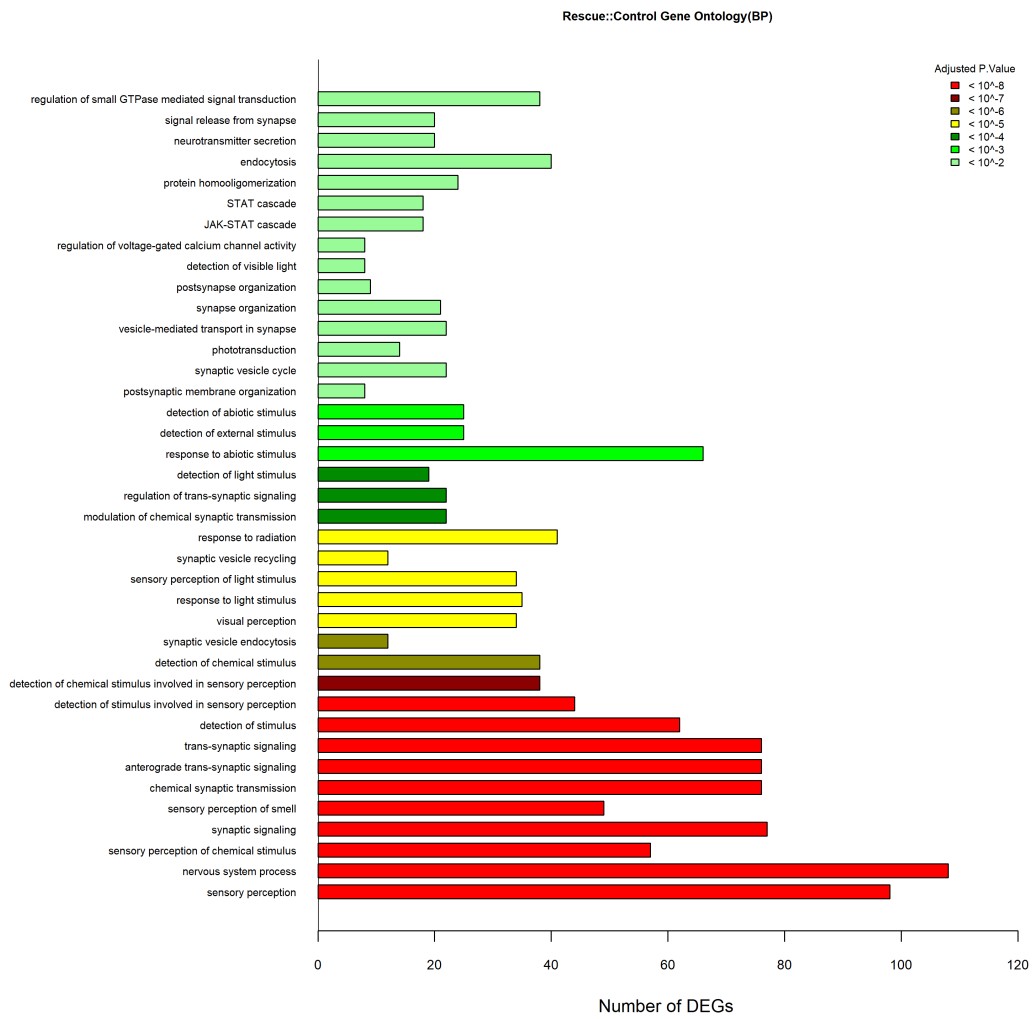

**Figure 7 GO Analyses Rescue Vs. Control.** Gene Ontology biological process enrichment results ($P <$ 0.01) from *lepa* rescue compared to control treatments representing $n = 39$ terms. Color scale reflects adjusted *p*.value; the *x*-axis denotes the number of DEGs that map to each enriched term.

(*jak2a*, *jak2b*), forkhead box O1 (*foxo1b*), AMP-activated protein kinase (*prkaa2*), acetyl-CoA carboxylase beta (*acacb*), calcium/calmodulin kinase (*camkk2*), mitogen-activated protein kinase kinase kinase 12 (*map3k12*), and suppressors of cytokine signaling (*socs2*, *socs5b*, *socs9*) (Fig. 4B). Consequently, KEGG phosphatidylinositol signaling (Fig. 3) and GO biological process JAK/STAT cascade (Fig. 7) are enriched in *lepa* rescue zebrafish. In addition to *prkaa2* and *acacb*, lipolytic factors also respond to *lepa* rescue (but not knockdown) including pantothenate kinase 2 (*pank2*) which mediates a critical step of coenzyme-A biosynthesis, lipoprotein lipase (*lpl*), diacylglycerol lipase (*dagla*, *daglb*), perilipin-1 (*plin1*), carnitine O-octanoyltransferase (*crot*), and muscle carnitine palmitoyltransferase 1B (*cpt1b*) (Fig. 4B).

Although lists of differentially expressed genes are a useful outcome of these studies, perhaps the most relevant results come from GO (gene ontology) and pathway analyses (e.g.,

KEGG). The goal of these analyses is to uncover functions/pathways that are statistically over/underrepresented in the treatment. Once a pathway is identified, all genes within the pathway are legitimate targets for downstream hypothesis testing, regardless of whether each gene in the pathway is differentially expressed (*Khatri, Sirota & Butte, 2012*). Notch signaling is differentially regulated between *lepa* knockdown and rescue treatments (Fig. 4A; Table 4). Notch signaling is a conserved juxtacrine signaling pathway that, in addition to many other roles, directs neuron cell fate (*Wakeham et al., 1997*). Notch signaling, neurogenesis, and neural differentiation respond to *lepa* knockdown (Fig. 4A; Fig. S3). GO enrichment pertaining to synaptic signaling, sensory perception, and neurotransmitter secretion is enriched in both *lepa* knockdown and rescue treatments compared to controls Figs. 5 and 7. *Ob/Ob* and *Db/Db* rodents have impaired nerve fiber extensions in arcuate nucleus (*Bouret, Draper & Simerly, 2004*). Similarly, *lepa* and *LEPR* zebrafish knockdown have thinning of spinal nerves along with malformations of sensory organs (*Liu et al., 2012*) indicating that there may be conserved roles between the embryonic zebrafish and human leptin signaling pathways tied to neurogenesis and synapse organization. This pathway analysis may also be less susceptible to gene:gene variation among replicates. Validation of expression data for 10 genes between the microarray and control groups is generally good, with 8/10 genes matching in the direction of change (e.g., up or down both in microarray and qPCR data, Fig. S4). However, agreement between the two methods for the rescue vs. control comparison is poor (1 of 5). We speculate that this treatment is inherently more complex (both knockdown with MO and rescue with RX leptin), and thus the response more variable. Other transcriptomics studies also report relatively low rates of agreement between transcriptomic and qPCR data (*Cui et al., 2014*).

Other laboratories investigated transcriptomic response to leptin manipulation in non-mammals. Denver's laboratory measured individual gene response and transcriptome response to leptin incubation in *Xenopus* brain preparations (*Cui et al., 2014*). They noted strong response of *socs* genes, similar to this study (Fig. 4B). Recently, Borski's laboratory published results of an *ex vivo* transcriptomic study on adult tilapia pituitary tissue treated with tilapia Leptin-a in culture (*Douros et al., 2018*). The overwhelming effect was increased carbohydrate metabolism, consistent with a leptin knockout in zebrafish (*Michel et al., 2016*). Several aspects of our study are consistent with *Douros et al. (2018)*, such as protein kinase B (*akt3*), Janus kinase 2 (*jak2a*, *jak2b*), insulin receptor substrate (*LOC794738*, *LOC100537326*), AMP-activated protein kinase (*prkaa2*), acetyl-CoA carboxylase beta (*acacb*) (Fig. 4B), and the general stimulatory effects on ribosome assembly and function (Table 4). Differences between the two studies likely reflect developmental window (embryos vs. adults), and tissue (whole embryo vs pituitary and liver); it is not surprising that leptin may have different effects on the transcriptome of developing whole embryos versus specific tissues of adults, even in the same species.

Phototransduction genes elicit the same response to *lepa* knockdown and rescue treatments compared to controls suggesting this effect is not directly correlated with Leptin-a signaling (Fig. 4A; Table 4). Correlations between the *lepa* knockdown dataset and unrelated morpholinos suggest that there may be a phototransduction artifact of morpholinos which is often manifested as reduced eye size and opsin expression. Having
acknowledged these possibilities, it is also important to note that these artifacts are typically seen at much higher morpholino dose/embryo than used in this study. Although we assert that our results generally reflect the effects of *lepa* knockdown in zebrafish embryos, it is also likely that the general effects of morpholino oligonucleotides are reflected in the data. Eukaryotic gene expression is regulated at multiple levels including transcriptional machinery and epigenetic factors, mRNA processing and transport, and post-translational modifications. The transcriptome represents the complete spectrum of RNA species that are present at one reference point in time. Expression microarrays are molecular technologies adapted to qualitatively and semi-quantitatively describe transcribed regions of the genome using a single experiment. Structurally, antisense morpholino oligonucleotides (MO) are short ∼25 mer, nucleic acid analogues with alternative morpholine rings in place of the native deoxyribose and ribose moieties present in DNA and RNA, respectively (*Summerton & Weller, 1997*). Uncharged phosphodiamorate linkages between morpholino nucleotides juxtapose the negatively charged phosphodiester backbones of native DNA/RNA. These structural rearrangements confer resistance to RNase H nucleolytic cleavage while the uncharged MO backbone reduces off-target electrostatic interactions during delivery and diffusion (*Summerton, 1999*). MOs are generally directed against the translation initiation site of a sense strand mRNA as described here. Alternatively, mRNA transport, maturation, and processing can be manipulated by directing MOs against splice boundaries of precursor mRNAs (*Kloosterman et al., 2007*). Gene knockdown is catalyzed by complementary base-pairing between antisense MO and target mRNA. Translation-blocking MO:mRNA hybrids sterically exclude the ribosome from executing translation which leads to reduced target protein synthesis and/or impaired processing of target RNA (*Bill et al., 2009*; *Summerton, 2007*). However, MOs generally have a temporal range of effectiveness. Cytological concentrations of MOs are reduced with every subsequent cell division which restricts most MO applications to early life stages (*Nasevicius & Ekker, 2000*).

Morpholinos have mistargeting potentials associated with *tp53* expression programs, neuron death, and hindbrain development; p53 morpholino co-morphants alleviate most of these artifacts (*Gerety & Wilkinson, 2011*; *Robu et al., 2007*). Zebrafish *lepa* knockdown share morphological features with zebrafish morphants from other target genes, such as pericardial edema, enlarged yolk sac, reduced eye and otolith size, as well as tail curvature (*Bagci et al., 2015*; *Kok et al., 2015*; *Kwon, 2016*; *Liu et al., 2012*; *Pham et al., 2007*; *Robu et al., 2007*). It is plausible that the formation of double stranded morpholino:RNA hybrids may enhance the RNA-induced silencing complex as ncRNA metabolism and RNA processing are among the top-ranked GO terms in *lepa* knockdown zebrafish (Fig. 5) which include Smg-5 nonsense mediated mRNA decay factor (*smg5*), Dicer-1 (*dicer1*), protein argonaute-1-like (*LOC570775*), and putative ATP-dependent RNA helicase DHX33-like (*LOC796505*) (Fig. 4B).

Finally, we understand that the interaction among the morpholino and recombinant protein is/are likely complex. The degree to which the knockdown have reduced expression, and the degree to which the recombinant leptin replaces what is reduced by morpholinos is unlikely a perfect match. This undoubtedly contributes to different pathways identified in each of the comparisons.

## CONCLUSIONS

These microarray data describe cell signaling pathways and gene targets regulated by embryonic zebrafish Leptin-a. Differentially expressed genes from the *lepa* rescue zebrafish embryos influence expression of genes that participate in central endocrine and phosphatidylinositol signaling pathways which agrees with the function of leptin signaling in mammals. Regulators of metabolism respond to *lepa* rescue that agree with *LEP/AMPK/ACC* and *LEP/PI3K/AKT* cascades in human (*Minokoshi et al., 2002*) (Fig. 4B). There are also markers consistent with increased central leptin signaling in the CNS among *lepa* rescue zebrafish which includes cocaine-and-amphetamine regulated transcript like-1 (*cart1*; 0.99 log2 fold change) and agouti-related peptide-2 (*agrp2*; -1.21 log2 fold change). Although we did not identify differentially expressed members of the STAT transcription factor family, Janus kinase 2 (*jak2a, jak2b*), insulin receptor substrate (*LOC794738, LOC100537326*), insulin receptor a (*insra*), Ser/Thr protein kinase B (*akt3*), PI3K (*pik3r1*), and forkhead box O1 (*foxo1b*) are differentially expressed in *lepa* rescue zebrafish which aligns with coregulation of the central PI3K/Akt/FoxO1 cascade by leptin and insulin receptor signaling in mammals (*Xu et al., 2005*). Taken together, our results propose that cell signaling pathways regulated by the A-type zebrafish leptin paralog align with human and rodent leptins including *LEP/AMPK/ACC* and *LEP/PI3K/AKT* but functional assays using fish leptin signaling knockouts are needed to validate these gene expression data.

### Funding

The experiments in this article were supported by NIH 2R15DK079282-02 to Richard Londraville and Qin Liu and The Choose Ohio First Bioinformatics Scholarship Program provided by the Ohio Board of Reagents to Matthew Tuttle and Mark Dalman. The funders had no role in study design, data collection and analysis, decision to publish, or preparation of the manuscript.

### Grant Disclosures

The following grant information was disclosed by the authors:
NIH: 2R15DK079282-02.
Choose Ohio First Bioinformatics Scholarship Program.

### Competing Interests

The authors declare there are no competing interests.

### Author Contributions

- Matthew Tuttle conceived and designed the experiments, performed the experiments, analyzed the data, contributed reagents/materials/analysis tools, prepared figures and/or tables, authored or reviewed drafts of the paper, approved the final draft.

- Mark R Dalman conceived and designed the experiments, performed the experiments, approved the final draft.
- Qin Liu conceived and designed the experiments, performed the experiments, contributed reagents/materials/analysis tools, approved the final draft.
- Richard L. Londraville conceived and designed the experiments, contributed reagents/materials/analysis tools, authored or reviewed drafts of the paper, approved the final draft.

## Animal Ethics

The following information was supplied relating to ethical approvals (i.e., approving body and any reference numbers):

The University of Akron Institutional Animal Care and Use Committee provided full approval for these experiments (08-6B).

## Data Availability

Data are available at EBI, accession ID# E-MTAB-6548 https://www.ebi.ac.uk/arrayexpress/experiments/E-MTAB-6548/.

## Supplemental Information

Supplemental information for this article can be found online at http://dx.doi.org/10.7717/peerj.6848#supplemental-information.

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
