# Peer review of "Leptin-a mediates transcription of genes that participate in central endocrine and phosphatidylinositol signaling pathways in 72-hour embryonic zebrafish (Danio rerio)"

_PeerJ, doi:10.7717/peerj.6848_

## Round 0.1 · original submission · Major Revisions

Thank you for submitting your work to PeerJ. Based on feedback from three reviewers and my own reading of your manuscript I have decided on the recommendation of Major Revision and invite you to respond to the suggestions of each reviewer and resubmit your manuscript.

All three reviewers were positive about the contributions of your study. While I ask that you respond to all reviewer comments, there are several areas that appear to be most significant:

1. The need for a more clear description of the gene expression dataset, discussion of how it compares to previously published studies and the need for confirmation by another method such as quantitative PCR. There are a number of comments about how data are presented in figures 1-4.

2. The recommendation that a description of the earlier morpholino work be added to the paper’s introduction, concern that knockdown is not confirmed by western or mass spectrometry and whether control injections were done.

3. Concerns about the conclusion that leptin affects phosphatidylinositol signaling.

I also have a few suggestions that I invite you to address in your revision:

4. Are the microarray biological replicates from separate injection experiments, or were all embryos collected from the same injection experiment? This could be defined in the methods.

5. I would suggest explaining the KEGG pathway analysis and what this does in the table 1 legend and in the Methods. Is there a visual way to present the data in table 1?

6. The legend for the volcano plots showing DEG (false and true) could more clearly state that these are differentially expressed and non-differentially expressed genes. You state this in the figure description, but it takes some interpretation to figure out what the legend is showing.

7. Two reviewers have comments about the morpholino experiments, including what controls were used and how knockdown was confirmed. I think it would also be helpful to include some images of typical injected embryos so readers could assess whether there are signs of general toxicity from the injections. Could you also explain why the 72 hpf timepoint was selected for the microarray analysis?

8. Please consider adding a table of genes with the greatest change in expression between treatments. Genes of interest are named in the paper, but it may be helpful for readers to see the top 25-50 genes.

9. Many of your italicized words bump up against the next word in your text.

10. I would add an x-axis label to the graph in figure 4.

11. You make a great point about MOs non-specifically affecting eye. This is likely true, so it does not seem that figure 5 is needed as it is not related to leptin function. You might also want to remove the mention of phototransduction gene knockdown in the abstract if this may be a non-specific effect of the morpholinos.

·

Basic reporting

no comment

Experimental design

no comment

Validity of the findings

Please also see comments point c but in my opinion, the core findings of this study could be replicated in a biological replicate and a qPCR setting to increase validity.

Additional comments

This paper on leptin A mediated effects in Zebrafish development is a very timely and important study on the effect of leptin in a non-mammalian vertebrate as the role of leptin is not entirely clear in the non-mammalian species thus far looked at.

Specific points

1) line 119 Daneau -> Danieau

2) lines 114-124 - 2nL of 0.4mM morpholino was injected or a 1:1 solution of 0.4mM morpholino with rx leptin - also 2nL (i.e. half the concentration) or 4nL (i.e. a larger injection)?

3) As far as figure references in text are concerned I would find it helpful to reference 2a and 2b seperately as opposed to all references leading to figure 2 as a whole.

4) lines 187-198 - The heatmap contains a network of genes that follow similar trends - what exactly do you mean? I think this paragraph could benefit from some further explanation at least for non-microarray afficionados like me. From what I understand the most interesting samples are samples which show a higher intensity in morphants compared to WT (upregulation) which is reversed by the rescue (i.e. RS intensity comparable to WT) which looks like the first cluster Figure 2a and conversely samples that have a lower intensity in MO vs WT which is "reversed" in RS, i.e. again comparable intensity between RS and WT (cluster 2). Is this what you mean by "follow similar trends"? Also, what do you define as "reciprocal trends" and specifically which genes are you talking about regarding the "delta/notch" pathway and the "phototransduction pathway"?

5) lines 199-206 supplemental figure 2 (line 195) is referenced before supplemental figure 1 (line 201) - could be reversed or this paragraph could go before the heatmaps?

6) Am I missing something or is Figure 2b not referenced in the text? I think this is a highly interesting figure. How did you classify "leptin signaling genes"? What is the overlap of leptin responsive genes in zebrafish compared to other organisms? I think a very interesting analysis would be to compare overlap between a) rx zf leptin on zf sample (this study) with an in silico analysis of previously published mamalian microarrays/RNAseq of leptin responsive cells (the recent leptin responsive neuron paper from Martin Myers comes to mind) and b) with the recent microarray published by Bob Denver on xenopus leptin treatment. Are there common signalling modalities?

7) Why use one way of expressing the data in Figure 3 (including # of DEGs, p-value and gene ratio) and another to express the data for figure 4 (excluding gene ratio)? I believe that it would be easier for the reader to express the data in a consistent manner.

8) line 276/277 - please see the 2017 paper by Bob Denver on leptin treatment of Xenopus as well as a 2018 paper by Borski on tilapia.

9) lines 304-324 - You are listing a bunch of leptin target which are differentially regulated in lepa rescue but not knockdown and conclude that therefore leptin a mediates the JAK/STAT... cascades. I am not sure I follow the argument. Leptin morphants would have reduced leptin signalling which should be rescued on co-administration of rx leptin. On the example of SOCS 2, 5b and 9 - figure 2 shows that there is no or little difference between WT and MO but a reduction during rescue. However, exposure of the animal to the LepR agonist should induce SOCS (admittedly 3a/b). Why then would there be a reduction in SOCS upon co-injection of leptin and leptin morpholino and no change upon morpholino injection? While I have not carefully looked at the other targets mentioned I am concerned about the change specific to the rescue.

10) lines 356-368 - You mention side-effects of morpholino treatment, shared obsevations in leptin a morphants and conclude the argument with expression changes. What do the expression changes (ncRNA metabolism, RNA processing, mRNA decay) mean? Are these expresion changes specific to morpholino treatments? If so, is mRNA decay for example upregulated in various morphants and upregulated in these morphants as well? An expansion of this discussion would be helpful as the connection of the mentioned expression changes to the morphological fetures discussed is unclear.

General comment

I think this is a very good paper and am very interested in the results. However, personally I think the results could benefit from a clearer presentation. The authors present data on leptin knockdown compared to it's rescue (incomplete or with potential induction as the leptin concentration is not titrated) compared to WT. However, what I am missing is

a) a clear analysis as to the direction of changes. There are points where the authors discuss these such as phototransduction which appears to be differentially regulated in the same direction (up or down?) in both morphants and rescued animals compared to control. But what about genes that to me would constitute the most interesting - down or upregulated in morphants which is partially or fully mitigated in the rescue (i.e. 3 fold up in morphant compared to WT; no change, less of a change or even a reduction in the rescue compared to WT and consequent significant difference between rescue and morphant or not)? I do not see much discussion on these - why? Microarrays are semi-quantitative.

b) (also covered in point 6) What about a compare and contrast to other similar experiments? In the discussion there is a comparison to the dataset by Borski. What about Denver's dataset and possibly some mammalian datasets? While I understand that the specific treatments are different, Denver's dataset is also during development for example. Overlap of DEGs and lack thereof could be telling. The core leptin regulated immediate early genes likely overlap?

c) The core genes found in this study and also mentioned in the abstract appear to me to be hidden in the lists and GO/KEGG terms. This could be clearer, maybe with a figure of expression changes in the different treatments dedicated to these? This is the case in the expression heatmap of figure 2, but since these are taken as the key finding in this study I would find it helpful if they were highlighted in some way. Further, the core genes could warrant to be checked by qPCR as the microarray results should be replicatable in a qPCR setting.

·

Basic reporting

This was a well written manuscript by a group that has deep (if not seminal) roots in teleost leptin research. As such, they offer novel insight into how leptin affects a teleost transcriptome. Granted that the current experiment pertains specifically to whole embryonic danio (as acknowledged), the experiment sets the stage to further explore lepa function is adult fish organs. Their findings provide specific and categorical connections between lepa and gene activity, and will likely help guide future research in the elusive topic of fish leptin(s) by.

Experimental design

This is original, primary work to my knowledge, addressing a relevant research topic that has received increasing attention over the last decade. As in past work, this group provides a rare comparative perspective that puts fish leptin function in a context with that of other vertebrates. The general design seems sound in their directness (transcriptome of KO and Rescue groups compared to a Control group), although the techniques used are beyond this reviewer’s personal experience. That said, the methods are established and the author provides references. Specific concerns or questions are listed below.

Validity of the findings

The group openly addresses limitations of their methods or conclusions; however, this reviewer defers to assessment of the stats to someone with experience in this technique.

Additional comments

Intro:
Background info about leptin function in vertebrates generally, and fish specifically, is provided. Brief description of the experimental design good, with groups clearly defined, (KO, rescue and wild type), and their comparative value within the experiment. Hypothesis and goal clearly stated. A reader unfamiliar with this type of KO might appreciate a few details about how the technique works; this information is included more so in the Discussion, but might be better moved to the Intro. Similar brief explanation of GOs and KEGG Pathways in the Intro might help the reader follow the data.

MM:
Treatments are described in adequate detail.

Line 121. Microinjection (of MO): is there a justification/reference for the MO and lepa concentrations? I see a volume for the MO, but not the leptin treatment (1:1 = double volume of MO alone?). Is it possible to verify the extent to which lepa is KO’d with this method? (i.e. with gene KO, it can be confirmed by PCR. Perhaps southern blot for lepa expression?) Sham injection to controls? (Please consider if any of these need to be addressed)

125. Does 72 hpf coincide with a particular developmental stage? As I read it, each treatment n=4 x5 pooled embryos = 20 embryos (the numbers a little confusing). If so, this seems adequate.

153. Comparisons are nicely presented. DEG criteria is explicit and contains reference.

Results:
3.1 I suggest that Fig 1 be labelled A and B1-B3, or in such a way as to make this section easier to follow in parallel with the data presented. It is not clear which figure the # of DEGs refers to, and more detailed figure labeling and specific references would help clear this up, especially since the #s mentioned are not apparent in the graphs/diagram. This should be easy to clarify.

176. Clarify “top-5 ranked genes.” E.g. upregulated genes? DEGs? Total expression? Otherwise, the various up/downregulated genes/classes of note are well described.

187-198. This section refers to the two heatmaps, and Figure 2 is mentioned 4 times in one paragraph. This section would again benefit from an A/B distinction in Figure 2.

204 Good data check of date vs group.

208. Again, a brief description of GO would be nice, as well as what ‘enriched biologically processing terms’ means, since these are key to the analysis. The reader pieces it together eventually, but no reason not to provide a quick explanation. The terms listed seem somewhat randomly chosen from the list on the figure – maybe these are just the more interesting functions.

224. Need a figure reference somewhere in this paragraph. Presentation is otherwise consistent with that of previous data.

239. Suggestion to use either ‘wild type’ or ‘control’ throughout text to make it easier to skim for information.

250. Again, a brief description of KEGG pathway would be nice.

Discussion:
275. First study of its kind in non-mammals = great.

278-290. Nice description of MO that would be beneficial earlier in the paper (either move or include in intro).

297. Confused: Why would lepa morpholinos rescue lepa KO? Otherwise nice suggestion of developmentally distinct roles of leptin.

304-310. This section seems to address if the rescue group should look like the WT group (i.e. both expressing lepa), or else why they might in fact be different in some aspects, which could be very interesting. Should this be more explicitly discussed?

317. Kinase written 3 times

325. A welcome, but conservative/general, conclusion after much data is referenced. Consider if any further conclusion is warranted?

356-86. Be consistent between lepa and lepA. This is paragraph (along with the last paragraph in the Discussion) is a good acknowledgement of MO limitations, but that the group is confidence that the experiment functioned as expected. Could 365-68 be considered a validation of lepa KO using the MO technique (as mentioned earlier)?

Note: Fig 1 does not appear to be mentioned in the Discussion.

Conclusion:
We see the first explicit mention of metabolic and appetite (genes named, but not “appetite” specifically) regulation. These concepts should probably be mentioned in the Discussion prior to showing up in the conclusion since they are classically associated with leptin.

Reviewer 3 ·

Basic reporting

Tuttle et al. present work detailing the effects of lepa on embryonic zebrafish gene expression using a well designed combination of lepa knockdown and lepa rescue models and a microarray approach to transcriptomics. The language is largely clear and the literature review is thorough. The rationale for and structure of the article are both nicely laid out.

There are a few minor comments regarding the basic reporting.
Lines 28-32: I believe the authors are stating that leptin signaling in humans is known to regulated a number of factors whose homologues are regulated by lepa rescue in the present studies. The wording is a bit unclear though. Could you confirm this is the intended meaning of your sentence and possibly rephrase this to be more clear.
Line 57: For clarity should read "with ob/ob rodents, but not db/db".
Line 70-73: Restructure for clarity. There appears to be a comma splice here and the sentence is difficult to follow as currently punctuated.
Lines 98-99: The wording of the comparisons made is unclear. Use the phrasing from the methods as that was readily understandable.
Line 277: It seems, by the authors statements, that Douros et al. have previously performed transcriptomic analysis of altered leptin signaling in a non-mammal. This sentence should be rephrased to give the sense that this work adds to the comparative, species specific literature.
Line 346: Change vs. to verses and remove the parenthesis surrounding "even in the same species".

Additionally there are major concerns that, in my opinion, would have to be addressed before this manuscript is published.

Experimental design

Line 126: How many total embryos were used for each group? By my math the WT groups had 40 total embryos pooled to form 8 biological replicates?

Validity of the findings

Line 278: It is unclear from the data that lepa knockdown was actually achieved. Because the antisense morpholino blocks translation but not transcription the microarray data is useless to confirm the knockdown. Therefore, it is necessary to some validation that this knockdown AND rescue functionally ablate and enhance leptin protein levels in this context. A western blot, immunoassay, or mass spec approach is required.

Line 300-303: The statement that your findings highlight a role for leptin-a in phosphatidylinositol signaling pathways is not wholly warranted as there is no data suggesting that leptin actually regulates this pathway. Rather the data wholly support a role for leptin to regulate the gene transcription of proteins involved in this pathway. The distinction is critical as this manuscript offers no functional data regarding the physiologic outcomes of leptin signaling in this particular model.

---

## Round 0.2 · accepted · Accept

Thank you for your thorough consideration and response to the reviewers’ comments and your revised submission. I am happy to now accept your manuscript for publication in PeerJ. I have just two small suggestions for changes that you could make when submitting the draft during production.

1. Reviewer 1 was not clear if the microarray and qPCR experiments used independent samples. Your response to reviewers letter states that they did, so you might want to make this clearer in the qPCR methods section.

2. It looks like you need to add spaces between italicized words and the following word in the legends for Tables 1-3.

You will be given the option to make the reviews of your manuscript available to readers. Please consider doing so as this review record can be a great resource for readers of your paper and contributes to more transparent science.

Thank you again for your contribution.

# ·

Basic reporting

Since this re-submission addresses my concerns I limit myself to the "general comments to the author".

Experimental design

Since this re-submission addresses my concerns I limit myself to the "general comments to the author".

Validity of the findings

Since this re-submission addresses my concerns I limit myself to the "general comments to the author".

Additional comments

The authors have presented a revised manuscript that addresses the main points I had reserves about in the first version. The current manuscript is a good read for the leptin community and will hopefully engender research into some of the pathways identified as leptin targets.

One minor point - the qPCR and the microarray - were these the same RNA samples or was the qPCR carried out on a freshly prepared set of RNA? The authors address this in line 424 and it seems to be implicit in their argument that the qPCR data also constitutes a biological replicate. However this is not clear from the manuscript. If these were technical replicates (as it seems to me many other authors use), then I would be worried.

·

Basic reporting

See general comments.

Experimental design

See general comments.

Validity of the findings

See general comments.

Additional comments

I followed the author's rebuttal and am generally satisfied with the way my comments were addressed. As long as the editor is ok with the merit of the Liu 2012 paper that is heavily referenced for precedence in the current paper, then some of the major concerns about validation seem to be addressed. Otherwise, the manuscript reads more clearly now, in part by paraphrasing some of the material covered in Liu, as suggested (rather than requiring a parallel read of that paper), as well as by clarifying/explaining some of the technical aspects and pathways, etc., that are discussed in the current paper The figures have been completely revamped, I see, although I am not very familiar with the types of analyses in Fig 2 &3, so cannot comment on them; however, the other tables and figures look intuitive. In a nutshell, the paper now more clearly illustrates differentially expressed genes, and hence has potential to inform readers of possible pathways affected by leptin, which has potential to drive new directions of leptin research.

Reviewer 3 ·

Basic reporting

No change from previous review.

Experimental design

No change from previous review.

Validity of the findings

No change from previous review.